# Video-based Computer-aided Laparoscopic Bleeding Management: a Space-time Memory Neural Network with Positional Encoding and Adversarial Domain Adaptation

**Navid Rabbani**[1]                                              navid_rabbani@yahoo.com
**Callyane Seve**[2]                                             callyaneseve@gmail.com
**Nicolas Bourdel**[2]                          nbourdel@chu-clermontferrand.fr
**Adrien Bartoli**[1]                                            adrien.bartoli@gmail.com

[1] *EnCoV, Institut Pascal, UMR 6602, CNRS/UCA/CHU, Clermont-Ferrand, France*

[2] *Department of Gynecological surgery, CHU Clermont-Ferrand, France*

## Abstract

One of the main challenges in laparoscopic procedures is handling intraoperative bleeding. We propose video-based Computer-aided Laparoscopic Bleeding Management (CALBM) for early detection and management of intraoperative bleeding. Our system performs the online video-based segmentation of bleeding sources and displays them to the surgeon. It hinges on an improved space-time memory network, which we train from real and semi-synthetic data, using adversarial domain adaptation. Our system improves the IoU and F-Score from 69.97% to 73.40% and 50.23% to 58.09% in comparison to the baseline space-time memory network. It is far better than the prior CALBM systems based on still images, which we reimplemented with DeepLabV3+, reaching an IoU and F-Score of 65.86% and 43.19%. The improvement is also supported by user evaluation.

**Keywords:** Bleeding Segmentation, Computer Assisted Intervention, Laparoscopic Surgery, Video Semantic Segmentation, Adversarial Domain Adaptation.

## 1. Introduction

Laparoscopy is a successful mini-invasive surgical approach. However, in spite of the numerous technological developments, it still presents unresolved challenges. One of the main challenges is handling intraoperative hemorrhage. For instance, as shown by Grant-Orser et al. (2014), 2.3% of laparoscopic hysterectomies end up with substantial blood loss, greater than 1000 ml. The consequences of hemorrhage range from an increased surgery duration and hospital stay to death, as reported in Mases et al. (2000). Intraoperative bleeding is generally handled laparoscopically or by conversion to open surgery. An early spotting of bleeding may improve its handling and reduce its impact. For that, Computer Assisted Surgery may play a fundamental role. A possible solution for decreasing the time needed for management of intraoperative haemorrhage is to use a real-time Computer-Assisted Laparoscopic Bleeding Management (CALBM) system. A CALBM system would directly indicate the bleeding sources in the laparoscopic images and can be anticipated to decrease blood loss, postoperative complications and number of conversions (Cheung et al., 2013; Fuks et al., 2016; Buell et al., 2009). In the longer term, it may as well increase the number of laparoscopic procedures, for it might give a reliable environment for procedures prone to intraoperative bleeding, usually performed by open surgery (Buell et al., 2009).

We cast CALBM as a semantic segmentation problem and propose a deep neural network to automatically segment the bleeding regions in laparoscopic videos. This is a challenging task, owing to the visual ambiguity between active bleeding, coagulated blood and some normal tissues. In addition, the bleeding regions have arbitrary and possibly complex shapes and are largely textureless. It is thus expected that temporal features may be required to achieve precise segmentation. This is supported by evidence observed in manual expert segmentation, where using a short video clip allows the expert to disambiguate a situation compared to using only a still image. The visual ambiguity and requirement for video make data collection complex and costly. It is thus fundamental to be able to exploit both supervised and semi-supervised learning. We observed that subjectivity occurred at times in bleeding labelling, owing to the ambiguity between running and steady blood, as well as neighbouring structures. Video mitigates this subjectivity but does not resolve it entirely. Another source of subjectivity is the typically faded edges of the bleeding areas.

We bring five main contributions. First, we introduce the first video-based CALBM system, which we concretely demonstrate in gynecologic laparoscopic surgery. Second, we propose an improved space-time network that uses a novel spatio-temporal positional encoding to explicitly capture time-dependent information. Third, we introduce a dataset with 949 labelled images and over 60 hours of unlabelled videos. Fourth, we train from both real and semi-synthetic data, using an extension of our network to achieve unsupervised domain adaptation. Fifth, we report an extensive experimental evaluation, comparing our networks to two baselines, the space-time memory network and DeepLabV3+. Standard segmentation metrics show that our networks outperform. User evaluation shows that they achieve a score close to manual ground-truth and far better than the baseline methods.

## 2. Related Works

**Bleeding detection in laparoscopy.** Existing research in CALBM is rather limited, in spite of the crucial importance of bleeding management. Jo et al. (2016) use a simple colour threshold in the CIELAB colour representation. Garcia-Martinez et al. (2017) use a similar approach with colour channel ratios, namely B/R and G/R. Okamoto et al. (2019) use a Support Vector Machine (SVM) classifier from combinations of RGB and HSV colour values. These three methods only consider the per-pixel colour information and ignore the valuable inter-pixel spatial information, obtaining poor results. Yamamoto et al. (2020, 2021) use neural networks designed for segmentation, namely the U-Net and YOLO-V3. By considering spatial information, these two methods outperform the three previous ones. However, they ignore the valuable inter-image temporal information. In contrast, our proposed network uses both spatial and temporal information.

**Bleeding detection in endoscopy.** Finding abnormalities such as bleeding sources is an active research area in gastrointestinal endoscopy (Tuba et al., 2017; Ghosh et al., 2018; Hajabdollahi et al., 2019; Li et al., 2019a). Gastrointestinal endoscopy and laparoscopy have strong differences, owing to the part of the anatomy they show. A gastrointestinal endoscopy image shows one organ and is typically more homogeneous than a laparoscopy image, which shows multiple ones. Nonetheless, the systems designed for bleeding detection in endoscopy may be inspiring for CALBM. Tuba et al. (2017) use handcrafted local features and an SVM classifier. Hajabdollahi et al. (2019) use a Multi-Layer Perceptron (MLP)

taking RGB patches. Ghosh et al. (2018) use a CNN SegNet. Li et al. (2019a) use a multi-stage attention U-Net. Similarly to the existing CALBM systems, some of these use the spatial information, but none use the temporal information, contrasting with the proposed approach which uses both. There also exist related methods towards the segmentation of organ parts other than bleeding. In particular, Casella et al. (2021) use 3D convolution layers to segment the inter-fetal membrane in endoscopic images while Kamnitsas et al. (2017) use unsupervised domain adaptation in brain lesion segmentation.

**Image and video segmentation.** There has been substantial research efforts recently in developing deep learning based methods for semantic segmentation. Video semantic segmentation (VSS) extends image semantic segmentation (ISS) to the spatio-temporal domain, using temporal information to facilitate temporal coherence. In ISS, the survey by Minaee et al. (2021) reported over 100 methods. Comparing the accuracy on the PASCAL VOC dataset (Everingham et al.) shows that the top-performing methods are EfficientNet+NAS-FPN (Zoph et al., 2020), DeepLabV3+ (Chen et al., 2018) and EMANet (Li et al., 2019b). We thus select DeepLabV3+ to form the baseline image-based CALBM system. In VSS, the survey by Wang et al. (2021) reported over 150 methods. The Space-Time Memory (STM) network (Oh et al., 2019) and its variants EGMN (Lu et al., 2020), LCM (Hu et al., 2021), and RMNet (Xie et al., 2021) show top performance on semi-automatic video object segmentation (SVOS). We thus select STM to form the baseline video-based CALBM system. However, STM does not fully exploit the powerful concept of temporal ordering, which we improve by including explicit positional encoding, and is not domain adaptation ready, which we improve by extending it in a domain adversarial manner.

## 3. Proposed Methods

We first introduce the baseline STM, then its proposed extension including positional encoding called STM-PE and its proposed extension handling domain adaptation called DA-STM-PE. The networks are shown in Figure 1.

### 3.1. The Space-time Memory Network

STM takes the current frame $I_t$ and the $T$ previous frames $\{I_{t-i}\}_{i=1:T}$ along with their segmentation masks $\{M_{t-i}\}_{i=1:T}$ as inputs. These masks are the network output at the previous temporal steps. It uses an attention mechanism, named space-time memory reader, to retrieve the spatio-temporal information. This exploits a memory value map based on the dot product attention between a query key map and the memory key map. The key and value pairs are produced by two different encoders, applicable to the query image and to the memory. The query encoder $\text{Enc}_Q$ takes the current frame as input and produces two feature maps, the key $k_Q$ and the value $v_Q$. It uses two parallel convolutional layers attached to the output of the backbone network. With these, the feature channel size of the backbone network is reduced by a factor of 8 for the key and 2 for the value. The memory encoder $\text{Enc}_M$ has a similar structure to the query encoder but takes both an image and a segmentation mask as inputs. These are concatenated along channel dimension to make a 4-channel tensor. Using an additional single channel filter, the 4-channel tensor is then fed

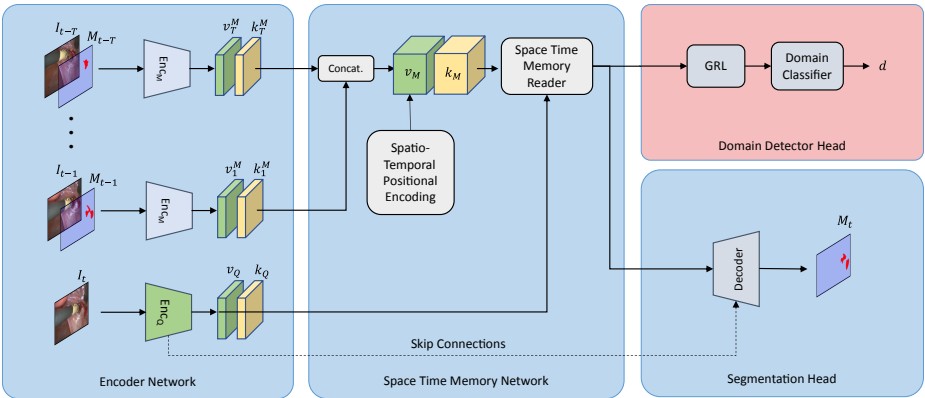

Figure 1: The proposed DA-STM-PE network extends STM with Positional Encoding (blue boxes) and adversarial unsupervised Domain Adaptation (red box).

to the backbone network. The backbone is a ResNet-50 and the features are derived from stage 4 of the network. The backbone weights are initialised from ImageNet training.

The keys and values from memory encoder are stacked along temporal dimension and produce the 4-dimensional tensors $k_M$ and $v_M$. We start with an empty mask as $M_0$ and follow the memory management rules of the baseline STM (Oh et al., 2019). The retrieved spatio-temporal tensor from the space-time memory reader is fed in a decoder to predict the segmentation mask $M_t$. The decoder uses multi-stage refinement blocks (Oh et al., 2018) to gradually upscale the feature map. At every stage the refinement block receives the output of the previous stage and the same resolution feature map from the query encoder from skip connections.

### 3.2. Extension to Positional Encoding

The extension made by STM-PE over STM is by explicitly capturing the spatio-temporal information into the value maps. In the attention mechanism of STM, this information is only implicitly captured. Concretely, we add the spatio-temporal information to the value maps $v_M$ using positional encoding. We use a positional encoding tensor PE, which has the same dimension as $v_M$ and can thus be directly concatenated. We use sine and cosine functions of different frequencies to produce PE, which is a spatio-temporal extension of positional encoding (Vaswani et al., 2017):

$$
PE(t,i,j,c) = \begin{cases}
\sin(c/10000^{i/T}) & \text{even}(i), \quad c < C/8 \\
\cos(c/10000^{(i-1)/T}) & \text{odd}(i), \quad c < C/8 \\
\sin(c/10000^{j/T}) & \text{even}(j), \quad C/8 \le c < C/4 \\
\cos(c/10000^{(j-1)/T}) & \text{odd}(j), \quad C/8 \le c < C/4 \\
\sin(c/10000^{t/T}) & \text{even}(t), \quad C/4 \le c < C/2 \\
\cos(c/10000^{(t-1)/T}) & \text{odd}(t), \quad C/4 \le c < C/2,
\end{cases} \tag{1}
$$

where $C$ is the number of channels and $t$, $i$, $j$ and $c$ are the indices for time, image $y$- and $x$-axes and channel.

### 3.3. Extension to Unsupervised Adversarial Domain Adaptation

While using the video improves temporal coherence and continuity compared to image segmentation, it makes training more challenging, owing to the need for a large amount of annotated video data. In particular, manually segmenting laparoscopic videos for a CALBM system would require prohibitive expert resources. We thus extend our STM-PE to DA-STM-PE, which trains using semi-synthetic videos by handling domain adaptation. We create semi-synthetic videos from annotated still images warped by random projective transforms, see Figure 2. The generated semi-synthetic videos used as training data and the real videos which we eventually wish to process have different distributions. This so-called domain shift can strongly harm performance if not handled properly. The purpose of domain adaptation is to address this domain shift. We implement it in DA-STM-PE via unsupervised adversarial domain adaptation (Ganin et al., 2016).

Unsupervised adversarial domain adaptation uses unannotated data from the destination domain, here the real videos, to adapt a network trained with the source domain, here the semi-synthetic videos. A general multi-layer segmenter network is, as shown in Figure 3, composed of a feature extractor, producing feature maps, and a segmenter, using the feature maps to produce the segmentation. Domain adaptation adapts the feature extractor by decreasing the $\mathcal{H}$-divergence between the feature maps produced for the source and destination domains. The $\mathcal{H}$-divergence between domains $\mathcal{S}$ and $\mathcal{T}$ is defined as:

$$d_{\mathcal{H}}(\mathcal{S}, \mathcal{T}) = 2(1 - \min_{h}[\text{err}_{\mathcal{S}}(h(X)) + \text{err}_{\mathcal{T}}(h(X))]),$$

where $h(X)$ is a domain classifier, specifically a binary function of the features $X$. The minimum term in $\mathcal{H}$-divergence definition is the domain classification error for the best domain classifier. By domain adaptation, the feature extractor network minimises the $\mathcal{H}$-divergence as:

$$f^* = \min_{f} d_{\mathcal{H}}(\mathcal{S}, \mathcal{T}) = \max_{f} \min_{h}[\text{err}_{\mathcal{S}}(h(X)) + \text{err}_{\mathcal{T}}(h(X))].$$

This is implemented with a Domain Adversarial Neural Network (DANN) (Ganin et al., 2016), using a Gradient Reversal Layer (GRL) and two loss functions, as shown in Figure 3. The proposed DA-STM-PE is a DANN which we construct from STM-PE. We thus add a domain detection head to STM-PE, as shown in Figure 1. This head uses a domain

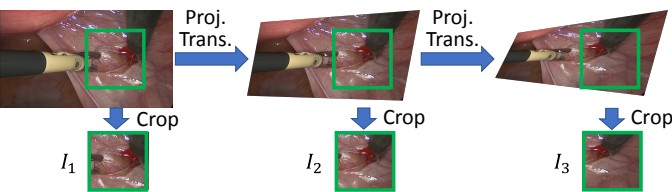

Figure 2: Semi-synthetic video clip synthesis from labelled still images.

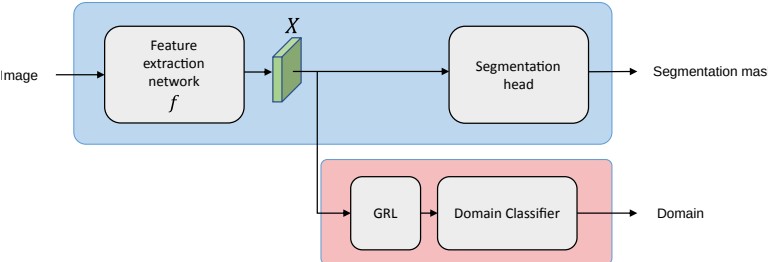

Figure 3: A segmenter network (blue box) and its extension as a Domain Adversarial Neural Network (DANN) (blue and red boxes).

classifier with a two-stage convolutional network following a global pooling layer. It classifies the domain of an input video as semi-synthetic or real. DA-STM-PE represents the first attempt ever to use domain adaptation with semi-synthetic videos generated from real still images for segmentation.

### 3.4. User Evaluation Method

In addition to the usual segmentation metrics, we evaluated the baseline and the proposed methods with a user score. For this purpose, we created an online questionnaire. The questionnaire shows 20 gynecologic laparoscopy images of various typical procedures. The images are shown side by side with their predicted bleeding masks. The questionnaire is then instantiated for each method and randomised. We sent these questionnaires out to gynecologic surgeons within and outside of our research team and asked them to score each bleeding mask between 1 and 5. The surgeons were given the following instructions: "If all bleeding sources were found with a reasonable extent of each area, then this is a very good result and scores 5. If all bleeding sources were missed, then this is a very bad result and scores 1. In between, the score depends on how complete the bleeding source detection is, and how useful it is to ensure coagulation." We also left the possibility to not score an image. Lastly, we collected the level of expertise, from junior to expert. We processed the responses to the questionnaires to derive an average score and confidence for each method.

## 4. Experimental Results

### 4.1. Datasets

We recorded over 60 hours of video from 96 gynecologic laparoscopic surgeries. All participants gave their written informed consent according to the approval of the ethical committee (IRB 2018-A03130-55). From these videos, 750 images with active bleeding were selected and annotated by junior surgeons under supervision of senior surgeons. The images were used to create 2-second semi-synthetic training videos by random rotation, shearing, zooming, translation and cropping. This was done on-the-fly during training. Consequently, we could virtually generate an infinite amount of these. In parallel, domain adaptation requires real video clips, albeit unlabelled. These were found as excerpts of the pool of unlabelled

videos, using random cuts, to produce 18560 such real unlabelled clips. From the number of training iterations, we estimated that around 160K semi-synthetic video clips were generated for training, the same number of video clips being randomly drawn from the pool of 18560 unlabelled clips used for domain adaptation. Ten 4-minute clips were randomly selected from the remaining videos to serve as test data, ensuring the absence of patient overlap between the training and test data. The surgeons then annotated 199 representative images, around 20 per video clip, to serve as ground-truth. From these, we constructed 199 test video clips, each consisting of 49 unlabelled consecutive frames followed by a 50th labelled frame at the end. The annotated dataset is publicly available at `http://igt.ip.uca.fr/~ab/code_and_datasets/datasets/bleeding_segmentation_v1p0.zip`.

## 4.2. Training

We downsampled all the images to $854 \times 480$ pixels. We trained from random image triplets from the semi-synthetic video clips and from the real videos with a maximum separation of 50 images. We used cross entropy as segmentation loss and the Adam optimiser. We used a constant learning rate of $10^{-6}$ and a GRL weight $\lambda = 0.001$. Training STM-PE and DA-STM-PE took 2 and 4 days on a PC with an NVIDIA Geforce 2080 Ti GPU.

## 4.3. Metric-based Evaluation

Evaluation on the test videos with the IoU and F-Score is shown in Table 1. We observe the same trend on both metrics. The video-based methods outperform the image-based DeepLabV3+. This clearly confirms the expectation that temporal information improves bleeding segmentation. Both proposed methods outperform the baseline STM, showing that spatio-temporal positional encoding improves performance. Finally, the proposed method with domain adaptation outperforms all other methods, showing that unsupervised domain adaptation compensates the domain shift of the semi-synthetic dataset.

Table 1: Metric-based evaluation results.

|             | DeepLabV3+ | STM    | STM-PE | DA-STM-PE |
|-------------|------------|--------|--------|-----------|
| **IoU**     | 65.86%     | 69.97% | 70.10% | 73.40%    |
| **F-Score** | 43.19%     | 50.23% | 51.18% | 58.09%    |

Table 2: User evaluation results. The scoring rate is at 100% for all methods.

|               | DeepLabV3+ | STM  | STM-PE | DA-STM-PE | GT   |
|---------------|------------|------|--------|-----------|------|
| Average score | 2.97       | 3.46 | 3.74   | 3.81      | 3.79 |
| Cohen's kappa | 0.12       | 0.25 | 0.20   | 0.14      | 0.28 |
| # responses   | 6          | 7    | 5      | 4         | 5    |

### 4.4. User Evaluation

We created five questionnaires, for the four methods and the ground-truth (GT). The questionnaires were sent to surgeons with various levels of expertise. Figure 4 shows sample data and Table 2 reports statistics. Generalised Cohen's kappa (Cohen, 1960; Gwet, 2014) is used to evaluate inter-rater agreement. This shows a slight to fair agreement between the expert responses, which is acceptable, given the subjective nature of bleeding segmentation. We observe the same trend as on the metric-based evaluation. In addition, the ground-truth score is very similar to DA-STM-PE, the best performing method, at about 3.8. This is because of the inherent ambiguities that exist in spotting bleeding and its extent, which leads to variability across the surgeons. The proposed methods perform far better than the baseline ones, scoring close to the ground-truth.

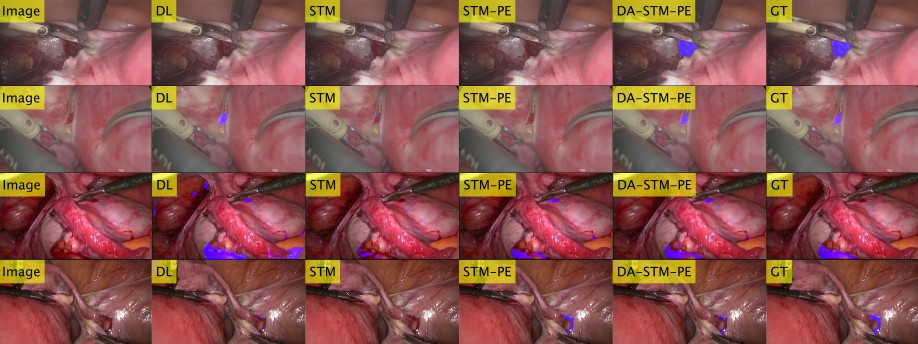

Figure 4: Sample segmentation masks from user evaluation. Left to right: original image, DeepLab3+ (DL), STM, STM-PE, DA-STM-PE and groundtruth (GT).

### 5. Conclusion

We have studied CALBM, for which we have proposed new STM networks. Our proposed STM-PE benefits from explicit spatio-temporal positional encoding, which the baseline STM lacks, and which improves performance. Our proposed DA-STM-PE is a further extension to achieve unsupervised adversarial domain adaptation. Using this new architecture, we have developed a framework for training video-based segmentation with semi-synthetic video data generated from still images. This is a valuable achievement for problems in which data annotation is extremely costly, and will thus be useful in other problems than CALBM. Our CALBM system is the first based on video-based segmentation and which exploits semi-supervised training. Our quantitative and user evaluations show the effectiveness of the proposed methods.

Our current work involves collecting additional surgeon responses for the user evaluation and testing our system in vivo. Our future work will involve extending our system to map the bleeding sources using SLAM and training for additional organs and procedures in laparoscopy and endoscopy, as well as finding an advanced evaluation metric.

## Acknowledgments

This research has been funded by Cancéropôle CLARA within the Proof-of-Concept project AIALO (2020-2023).

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

## Appendix A. Error Analysis

We discuss the failure modes of the proposed bleeding segmentation method and how they can downgrade the performance of a CALBM system. We manually inspected the segmentation masks from the proposed method and categorised the minor and major sources of failure as follows:

- **Minor boundary misalignment.** In this mode the bleeding segmentation masks visually look very similar to the ground-truth. There is nonetheless inevitably a slight degree of misalignment at the boundaries. For these, the IoU lies within 85% to 95%. Examples are shown in Figure 5. The misalignment is explained by the subjectivity in segmentation of the faded bleeding boundaries. This type of failure is minor and is not expected to impact a CALBM system performance.

- **Major boundary misalignment.** In this mode, the segmentation method is unable to segment the bleeding precisely. This type of failure is rare but can downgrade the visualisation. Examples are shown in Figure 6. A CALBM system should be able to cope with this failure, as it only happens at isolated frames, hence mitigating its clinical impact.

- **False positives.** In this mode, false bleeding sources are segmented. This is a rare event, occurring mainly owing to a confusion with steady blood, coagulated blood, blood vessels and similar reddish looking tissues. Examples are shown in Figure 7. This failure mode downgrades visualisation as flicker noise but does not downgrade the clinical reliability, as a CALBM system should be able to filter it out.

- **False negatives in early phase of bleeding.** False negatives are the most serious type of failure mode. These occur when the segmentation misses a bleeding source. We found a few cases of this failure mode. Examples are shown in Figure 8. These failures are temporarily and hence are not expect to seriously harm the system reliability.

- **False negatives in advanced phase of bleeding.** This failure mode can intensively harm the reliability of a CALBM system, as it would cause a bleeding source to be overlooked. We did not find any occurrence of this failure in our study.

The frequency of occurrence for each failure mode following our analysis is reported in Table 3. The perfect segmentation cases have an IoU greater than 95%.

Table 3: Occurrence of the different types of failure modes in our test dataset.

| Success and failure modes | Frequency |
| --- | --- |
| Perfect segmentation | 06.03% |
| Minor boundary misalignment | 48.24% |
| Major boundary misalignment | 26.63% |
| False positives | 14.07% |
| False negatives in early phase of bleeding | 05.02% |
| False negatives in advanced phase of bleeding | 00.00% |

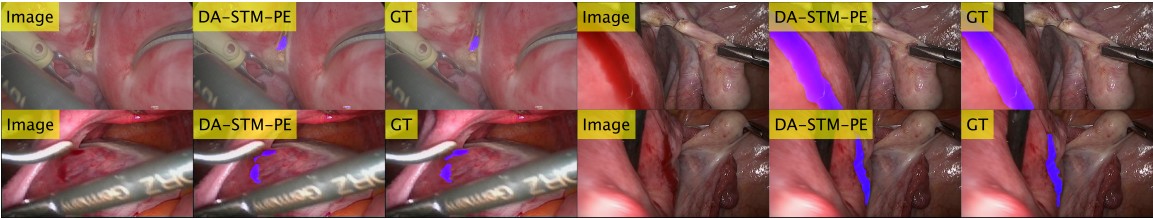

Figure 5: Sample segmentation masks with minor boundary misalignment.

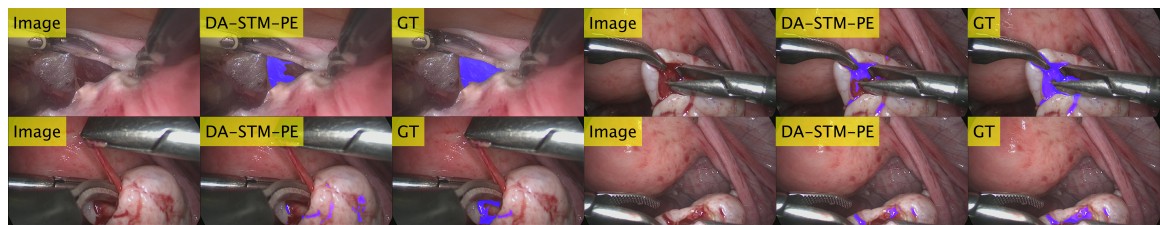

Figure 6: Sample segmentation masks with major boundary misalignment.

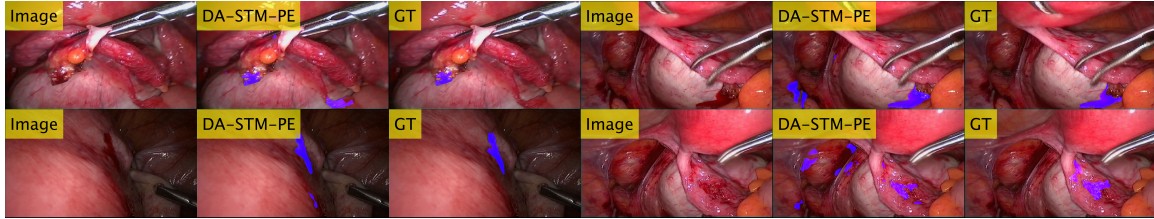

Figure 7: Sample segmentation masks with false positives.

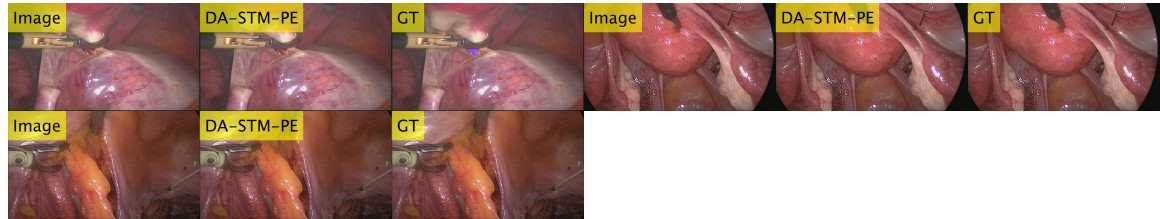

Figure 8: Sample segmentation masks with false negatives in early phase of bleeding.

