# OpenReview forum: "Video-based Computer-aided Laparoscopic Bleeding Management: a Space-time Memory Neural Network with Positional Encoding and Adversarial Domain Adaptation"
_MIDL.io/2022/Conference — MIDL 2022_

### Official Review · Reviewer_Zzpp · 2022-01-19

**Confidence:** 4
**Preliminary Rating:** 4
**Recommendation:** Poster

**Summary:**

The authors propose to modify Space-time Memory Network (STM) by adding the Spatio-temporal positional encoding and unsupervised Adversarial Domain Adaptation for bleeding segmentation in laparoscopic videos. The authors evaluate their method with their gynecologic laparoscopic video dataset and compare their proposed method with DeepLabV3+. The authors claim that their method can be applied to online bleeding segmentation for surgeons.

**Strengths:**

1. The idea of adding the Spatio-temporal positional encoding and unsupervised Adversarial Domain Adaptation to STM seems to be novel.

2. Besides metric-based evaluation, the authors also conduct user evaluations for their system.

3. The authors conduct a good literature review for prior work from the methodological development aspect.

**Weaknesses:**

1. Not enough details are provided for the dataset. This has a negative effect on the evaluation part of the paper. Please check on my detailed comments on this.

2. One big challenge for a Video-based Computer-aided Laparoscopic Bleeding Management system is to distinguish bleeding from blood. However, the paper did not address enough on this issue. Please check on my detailed comments on this.

3. The current method seems to be evaluated in an offline setting instead of an online setting. Please check on my detailed comments on this.

**Deanonymize Review:**

no

**Detailed Comments:**

For weakness point 1:

(1) It is very difficult for the reader to understand what is the detailed data split here. It seems that the author select some videos and annotate only part of them as the training set and test set. May I know how many videos are used for training/domain adaptation/testing? May I know how many images/video clips are used for training/domain adaptation/testing? How many semi-synthetic training videos are generated? A Table of the detailed dataset might be helpful for this part.

(2) If the dataset size is small and the test data is limited,  cross-validation should be done in the evaluation part of the paper.

For weakness point 2:

(1) Detecting bleeding is sometimes difficult due to the model failing to distinguish bleeding with blood presence. The authors claim their system can be used for online purposes. However, the authors did not share any analysis on how accurate the model can detect bleeding events.

(2) How many images in the training/test data have blood presence on them (Not bleeding)? Please share more details on this topic. For online applications, surgeons certainly do not want the model to misrecognize bleeding with blood presence.

For weakness point 3:

(1) The evaluation seems in offline mode. To capture different edge cases in online settings, why not annotate full video for bleeding segmentation. Will selecting short video clips and annotating 20 representative images per video clip for ground truth be enough to generate a representative dataset for this challenging problem?

(2) It will be great if the authors can provide us with error analysis. What are the edge cases where the model fails to detect bleeding? What are the edge cases where the model cannot segment the bleeding from images well? How these will affect the online application?

**Final Rating After The Rebuttal:**

5: Strong Accept

**Justification Of The Final Rating:**

I would like to thank the authors for their reply.

For weakness point 1, the authors revised the paper and share details about the data split. Although 199 representative images for testing are still kind of limited, I do agree that annotating the data is kind of challenging and the current test data is selected by surgeons. And the authors might not have enough time to perform cross-validation during this short rebuttal time.

For weakness point 2, the authors shared more details in the introduction section of the revision. The error analysis added in appendix A in the revision also helps to address this.

For weakness point 3, the authors provide a great error analysis in the revision. It helps to improve the paper's quality from the evaluation aspect. Although the dataset currently seems to be small, the results are promising and I believe the authors are in the right direction.

I believe the error analysis in the revision and the clarification of the data split improved the paper quality a lot. I would like to change my from "Weak Accept" to "Strong Accept".

**Paper Type:**

methodological development

**Questions To Address In The Rebuttal:**

The proposed system combines different model designs from the existing literature in a way that seems to be novel. However, the validation/application part of the paper is weak and needs revising. It is more a methodological development paper in my opinion.

1. From the evaluation aspect, the dataset section is not described clearly for the reader.  Please check on my comments for weakness point 1.

2. From the application aspect of the paper, the authors fail to address enough about bleeding vs blood presence. The performance of the model in online settings for application purposes becomes difficult to evaluate. Please check on my comments for weakness point 2.

3. The evaluation seems in offline mode. It seems not enough online scenarios are captured in their test dataset and no online performance analyses are given. Please check on my comments for weakness point 3.

**Special Issue:**

no

---

### Official Review · Reviewer_dDDV · 2022-01-22

**Confidence:** 5
**Preliminary Rating:** 3
**Recommendation:** Poster

**Summary:**

The paper proposed a space-time memory network that learns from real and semi-synthetic data, using adversarial domain adaptation for early detection and management of intraoperative bleeding during laparoscopic procedures. The paper compares and evaluated with SOTA methods and improves performance by a reasonable margin.

**Strengths:**

- the paper is reasonable well written and clearly presented
- the combination of known approaches is sound and the results show good improvements
- a new dataset is introduced comprising of 60 hours of videos including about 1000 labelled images
- a user evaluation is provided

**Weaknesses:**

- dataset probably not public
- methodological insights are limited, not going far beyond the addition of more input dimensions and more data with adversarial domain adaption
- no ablation study provided

**Deanonymize Review:**

no

**Detailed Comments:**


The contribution section states "we introduce a dataset of over 60 hours of videos including about 1000 labelled images.", which sounds like that this dataset will be provided with the paper but I cannot see a download link of project page.

since several user responses are available from the surveys, it could be considered to also evaluate inter-rater agreement.

Kamnitsas K, Baumgartner C, Ledig C, Newcombe V, Simpson J, Kane A, Menon D, Nori A, Criminisi A, Rueckert D, Glocker B. Unsupervised domain adaptation in brain lesion segmentation with adversarial networks. InInternational conference on information processing in medical imaging 2017 Jun 25 (pp. 597-609). Springer, Cham.
should be added to the discussion of domain adaption.

The results are still not great, thus I am wondering how relevant such a method would be in practice. Is IoU perhaps not relevant for the downstream task, or is perhaps missing a bleeding much worse than a false positive (the something like a F2 score could also be evaluated if the metric should be either biased towards false positives or false negatives)

**Final Rating After The Rebuttal:**

4: Weak Accept

**Justification Of The Final Rating:**

Thank you for addressing the points raised above and adding clarifications to the presented work. Though, my review is also not great or extensive so in dubio pro reo but I am really undecided here. Please weigh the others reviewers more.

**Paper Type:**

methodological development

**Questions To Address In The Rebuttal:**


Could the authors please comment on dataset availability and reproducibility?

Also please comment on relevance of the finding for the clinical practice and about the acceptable error in real-world applications.

**Special Issue:**

no

---

### Official Review · Reviewer_czLE · 2022-01-25

**Confidence:** 4
**Preliminary Rating:** 4
**Recommendation:** Poster

**Summary:**

The paper presents a Computer-aided Laparoscopic Bleeding Management (CALBM) for early detection and management of intraoperative bleeding that exploits temporal information through a space-time memory network with domain adapttion. Moreover, a novel dataset for laparoscopic bleeding detection has been introduced to test the proposed method.

**Strengths:**

- Despite the limited novelty of the work, it address a relevant problem for the medical society
- A novel dataset of about 1000 labelled images has been proposed
- The paper is well written and easy to read
- An Ablation study helps validate the efficacy of the proposed method

**Weaknesses:**

- Despite a dataset is presentend, the amount of frames is limited and the cliam of the authors is misleading
- The improvements to the space-time network are not clearly presented
- The literature review on the spatio-temporal methods is limited
- The purpose of the domain adaptation is not clearly introduced
- User evaluation is very limited and does not provide enough evidence to validate the results

**Deanonymize Review:**

no

**Final Rating After The Rebuttal:**

5: Strong Accept

**Justification Of The Final Rating:**

I would like to thank the authors for addressing my questions and improving their manuscript.
Despite the good quality of the paper, the amount of data for testing the method is still limited - this should be addressed in the future.

**Paper Type:**

both

**Questions To Address In The Rebuttal:**

- The claim of the authors could be misleading: "Third, we introduce a dataset of over 60 hours of videos including about 1000 labelled images." while the dataset contains only 1000 frames.
- "A real-time Computer-Assisted Laparoscopic Bleeding Management (CALBM) system, which would directly indicate the bleeding sources in the laparoscopic images, may dramatically aid the surgeon It would decrease blood loss, postoperative complications and number of conversions. In the longer term, it may as well increase the number of laparoscopic procedures, for it might give the surgeon a reliable environment for procedures prone to intraoperative bleeding, usually performed by open surgery." please provide some reference on these statements
- The author claim "Second, we propose an improved space-time memory network with spatio-temporal positional encoding added to value maps" please improve and clarify better where the improvents lies
- The user evaluation should be extended to a wider cohort. This quantativa evaluation cannot be used in the actual form. If it won't be possible to extend the cohort, please consider to remove it.
- Please consider to extend the literature review, despite to the tecniche of positional encoding, other work exploits the Spatio-Temporal information in laparoscopic video as in A shape-constraint adversarial framework with instance-normalized spatio-temporal features for inter-fetal membrane segmentation (Casella, Medical Image Analysis, 2021)
- In Figure 1, the labels that indicate the temporal position of the inputs do not agree with what is written in chapter 3.1
- In the description of the positional encoding scheme PE(t,i,j,c), all the variables should be declared and explained. k variable is not reported
- Please consider to include a graphical example of the synthetic data used, warpings and other elements like a table to summarise and better understand the domain adaptation scheme
- Consider to extend the description of Figure 3 defining acronym and type of image. Consider also to increase the font size of the labels in the image.


**Special Issue:**

no

---

### Meta-Review · Area_Chair_1HZD · 2022-02-18

**Recommendation:** Accept (Oral)
**Confidence:** 4

**Metareview:**

According to the reviewers, the quality of the paper improved alot during the rebuttal phase (2x strong accept, 1x weak accept). I also recommend acceptance of the paper. The authors explained the data set split in more detail and provided other very important information, that helps in reproducing the work. The method was evaluated with SOTA methods and performance was improved by a reasonable margin.

---

### Decision · Program_Chairs · 2022-02-28

Accept